# Diversity, Prevalence and Virulence of *Colletotrichum* Species Causing Anthracnose on Cassava Leaves in the Northern Region of Brazil

**DOI:** 10.3390/jof10060367

**Published:** 2024-05-21

**Authors:** Stella de C. S. Machado, Josiene S. Veloso, Marcos P. S. Câmara, Willie A. S. Vieira, Luis O. Viteri Jumbo, Raimundo Wagner S. Aguiar, Alex Sander R. Cangussu, Marcos V. Giongo, Cristiano B. Moraes, Fabricio S. Campos, Sabrina H. C. Araújo, Eugênio E. Oliveira, Gil R. dos Santos

**Affiliations:** 1Programa de Pós-Graduação em Produção Vegetal, Universidade Federal do Tocantins, Gurupi 77402-970, TO, Brazil; stella.machado@ifpa.edu.br; 2Departamento de Agronomia, Universidade Federal Rural de Pernambuco, Recife 52171-900, PE, Brazil; josieneveloso@yahoo.com.br (J.S.V.); marcos.camara@ufrpe.br (M.P.S.C.); 3Departamento de Fitopatologia, Universidade de Brasília (UnB), Brasília 70910-900, DF, Brazil; andersonvieira12@gmail.com; 4Programa de Pós-Graduação Ciências Florestais e Ambientais, Universidade Federal do Tocantins (UFT), Gurupi 77402-970, TO, Brazil; luis.viteri@mail.uft.edu.br (L.O.V.J.); giongo@uft.edu.br (M.V.G.); cbmoraes@uft.edu.br (C.B.M.); 5Programa de Pós-Graduação em Biotecnologia, Universidade Federal do Tocantins, Gurupi 77402-970, TO, Brazil; rwsa@uft.edu.br (R.W.S.A.); alexcangussu@uft.edu.br (A.S.R.C.); camposvet@gmail.com (F.S.C.); 6Departamento de Entomologia, Universidade Federal de Viçosa (UFV), Viçosa 36570-900, MG, Brazil; shcaraujo@gmail.com (S.H.C.A.); eugenio@ufv.br (E.E.O.)

**Keywords:** anthracnose, *Manihot esculenta*, variability, severity, phylogenetic analysis

## Abstract

Cassava (*Manihot esculenta* Crantz) is a staple crop widely cultivated by small farmers in tropical countries. However, despite the low level of technology required for its management, it can be affected by several diseases, with anthracnose as the main threat. There is little information about the main species of *Colletotrichum* that infect cassava in Brazil. Thus, the objective of this work was to study the diversity, prevalence and virulence of *Colletotrichum* species that cause anthracnose in cassava leaves in northern Brazil. Twenty municipalities of the Pará and Tocantins states were selected, and leaves with symptoms were collected in those locations. Pure cultures were isolated in the laboratory. Species were identified using phylogenetic analyses of multiple loci, and their pathogenicity, aggressivity and virulence levels were assessed. Our results showed the greatest diversity of *Colletotrichum* associated with anthracnose in cassava plants of the “Formosa” cultivar in the Tocantins and Pará states. We determined the presence of *Colletotrichum chrysophilum*, *C. truncatum*, *C. siamense*, *C. fructicola*, *C. plurivorum, C. musicola* and *C. karsti*, with *C. chrysophilum* as the most aggressive and virulent. Our findings provide accurate identifications of species of *Colletotrichum* causing anthracnose in cassava crops, which are of great relevance for cassava breeding programs (e.g., the search for genotypes with polygenic resistance since the pathogen is so diverse) and for developing anthracnose management strategies that can work efficiently against species complexes of *Colletotrichum*.

## 1. Introduction

Cassava (*Manihot esculenta* Crantz) is widely cultivated by small farmers in tropical countries in Africa, Asia and Latin América. Due to its easy cultivation and production of starch-rich roots, cassava is emerging as the most important tropical tuber crop [1,2]. According to the Food and Agriculture Organization of the United Nations, “cassava is very important in feeding half a billion people in the world” [3,4,5,6]. Brazil is the third-largest cassava producer in the world [7,8,9], with the State of Pará as the largest national producer. The State of Tocantins is also amongst the bigger State producers, showing the same trend of growing production and planted area [10]. In the northern region of Brazil, cultivation is predominantly low in technology, with losses in productivity due to several factors, including diseases caused by phytopathogens.

The cassava crop can be affected by several fungal diseases, and anthracnose is considered the most destructive foliar disease [11,12,13]. The symptoms are characterized by necrotic lesions of irregular shapes, distributed along the leaf blade. In young plants, it causes damage to the apical bud, causing the symptom known as “pointer drought” and, in more severe cases, the death of the plant. In adult mature plants, it can result in progressive defoliation [14,15]. In addition to attacking the leaves, anthracnose can affect the cassava plant in all its stages of development, causing canker on the stem and branches and also the death of the plant [16]. The greater incidence and severity of cassava anthracnose are found in more humid locations than in dry locations [14,17]. Under these conditions, losses caused by anthracnose can reach 90% [18]. The pathogen can be spread mainly by infected cuttings and can survive in stems and crop residues in the soil [19].

Historically, *Colletotrichum gloeosporioides* f.sp. *manihotis* has been reported as the causal agent of cassava anthracnose worldwide. During the past 20 years, molecular phylogeny has completely changed the classification of the genus *Colletotrichum*. Most of the early studies on anthracnose etiology were based on morphological characteristics, which is insufficient to correctly identify the diversity of *Colletotrichum* species associated with this disease [20,21]. Thus, accurately identifying the pathogenic species is critical to understanding the epidemiology of the disease and developing efficient control strategies.

Although there are some reports of *Colletotrichum* species in association with cultivated cassava [13,22,23], there is no information on the diversity, distribution and aggressiveness of the species causing anthracnose in northern Brazil. Thus, this study aimed to identify the species of the genus *Colletotrichum* associated with cassava in the region from southeastern Pará to northwest Tocantins, as well as determining the prevalence, aggressiveness and virulence of these species. Identifying the *Colletotrichum* will guide the management of anthracnose, leading to higher yields in this growing production area.

## 2. Material and Methods

### 2.1. Sampling, Isolation and Collection

From January to May 2018, symptomatic leaf samples were collected in 20 municipalities known as producers of cassava, on familiar farms in Pará (10) and Tocantins (10) States (Figure 1). In each municipality, five farms with *M. esculenta* in different stages of growth and maintained with family labor were sampled. In each farm, ten leaves of the upper third of the plants with typical symptoms of anthracnose were collected.

Infected tissue fragments obtained from the lesion margin, between the necrotic and healthy tissue, were disinfected in 70% ethanol for 30 s and 1% sodium hypochlorite for 1 min. The fragments were plated on potato dextrose agar (PDA) culture medium and incubated for five days at 25 °C with a photoperiod of 12 h. Isolates characteristic of the genus *Colletotrichum* Sutton [24] were isolated in a pure culture and stored in cryogenic tubes with sterile distilled water [25]. The isolates were deposited in the fungal collection of the Mycology Laboratory at Universidade Federal Rural de Pernambuco. Based on morphological characteristics (colony color, spore size and shape), isolates were selected randomly for level-specific identification through phylogenetic analysis.

### 2.2. DNA Extraction, PCR Amplification and DNA Sequencing

The *Colletotrichum* isolates were cultivated in PDA for 7 days at 27 °C, under a 12 h photoperiod in the Mycology Laboratory at Universidade Federal Rural de Pernambuco, PE, Brazil. Aerial mycelium was scraped from the colony surface, and genomic DNA was extracted following the cetyltrimethylammonium bromide (CTAB) protocol, with some modifications [26]. The DNA concentration was visually estimated on a 0.8% agarose gel.

The intergenic spacer between the 3′ end of the DNA lyase and the mating-type MAT1-2 (APN2/MAT-IGS) locus was initially amplified for all isolates to determine the species belonging to the Gloeosporioides complex. For those isolates in which this gene did not amplify, amplification of the partial region of the glyceraldehyde 3-phosphate dehydrogenase (GAPDH) gene was performed to identify if they belonged to other complexes and select the markers for multilocus analysis. The different haplotypes were identified using the DnaSP 4.0 software [27], and an isolate representing each haplotype was chosen at random and subjected to multilocus analysis. DNA lyase (APN2), glyceraldehyde-3-phosphate dehydrogenase-IGS (GAP2-IGS), β-tubulin (TUB2), calmodulin (CAL), chitin synthase 1 (CHS-1), histone H3 (HIS3) and actin (ACT) were amplified and sequenced for representative isolates according to the identity of the complexes. The primers used are listed in Appendix A.

The PCR conditions for APN2/MAT-IGS and APN2 constituted an initial denaturation step at 95 °C for 5 min, followed by 40 cycles at 95 °C for 30 s, 62 °C for 45 s, 72 °C for 1 min and a final extension at 72 °C for 10 min. The annealing temperature differed for each of the other genes: ACT and CAL—57 °C, GAPDH—55 °C, GAP2-IGS—58 °C, HIS3—52 °C and TUB2 and CHS-1—53 °C. PCR products were separated by electrophoresis on 1.5% agarose gel in 1.0X Tris-acetate-EDTA (TAE) buffer, photographed under UV light and purified by precipitation in ammonium acetate and ethanol. Sequencing of all loci was performed using ABI PRISM^®^ BigDye^®^ terminator v3 cycle sequencing kits (Applied Biosystems, Waltham, MA, USA), on the LABCEN/CCB sequencing platform at the Universidade Federal de Pernambuco, Recife, PE, Brazil.

### 2.3. Phylogenetic Analysis and Species Recognition

Nucleotide sequence quality analysis and consensus assembly were performed using the Staden Package v.2.0.0, 1998 [28]. All consensus sequences were compared to those from the NCBI nucleotide database using the BLAST algorithm [29], to confirm the taxonomic identities of the isolates. Sequences of *Colletotrichum* ex-type species were obtained from GenBank, where the sequences generated in this study were deposited (Appendix A). The alignment of the multiple sequences of each individual locus was estimated online using the G-INS-i strategy in the MAFFT 7 version [30,31], with default parameters for gap opening and extension and a 200PAM/κ = 2 nucleotide scoring matrix, and manually adjusted when necessary using MEGA6 [32].

Phylogenetic analyses were performed using Maximum Likelihood (ML) and Bayesian Inference (BI) methods for individual and concatenated genes. The ML and BI analyses were performed using RAXML-HCP2 v.7.0.4 (STAMATAKIS, 2014) and MrBayes v 3.2.1 [33], respectively, which were implemented in the CIPRES cluster (https://www.phylo.org/portal2/home.action (accessed on 13 June 2023)). ML analyses were performed with 1000 pseudo-replicates (-m GTRGAMMA -p 12345 -k -f a -N 1000 -x 12345) under the model GTRGAMMA.

Nucleotide substitution models were estimated in MrModeltest 2.3 [34], using the Akaike information criterion (AIC) for each locus individually. The concatenated matrix was partitioned with each locus with its referent nucleotide substitution model using SequenceMatrix v.1.8 [35]. Four Markov Monte Carlo chains (MCMCs) were conducted for 107 generations, and trees were sampled every 1000 generations. Convergence of all parameters was verified using Tracer v 1.5 [36], and the first 25% of generations were discarded as burn-in. FigTree version 1.4.3 [37] was used to visualize the phylogenetic tree.

Evolutionary lineages were recognized using the Genealogical Concordance Phylogenetic Species Recognition approach, as described in [38,39,40]. In this method, one lineage was considered to be independently evolved if it was strongly supported as monophyletic in the concatenated analysis and met at least one of the following criteria: one clade was concordant when it was present in most individual gene trees (i.e., minim in 4 of 7), and one clade was non-discordant when it was supported by BI (post probability 0.95) and ML (bootstrap 70%) analyses on a single gene tree and did not conflict with any other single genealogy at the same level of support.

### 2.4. Prevalence of Colletotrichum Species

The prevalence (P) of *Colletotrichum* species associated with anthracnose in *M. esculenta* was determined in the northern region of Brazil, separately, in the states of Pará and Tocantins, according to the municipality of collection. The prevalence was calculated with the following formula: P (%) = (Nx/Nt) × 100, where P = prevalence, Nx = number of isolates of the same species and Nt = total number of isolates [41].

### 2.5. Pathogenicity, Aggressiveness and Virulence

To verify the pathogenicity of *Colletotrichum* species, initially, the pathogenicity assay was performed on cassava leaves (Var. Formosa) close to 90 days old. Fungal inoculum was produced in Petri dishes contained PDA culture medium, under a 12 h photoperiod and 27 °C, until sporulation was confirmed. The recovered conidial suspension was adjusted to 2 × 10^6^ conidia mL^−1^ and sprayed on the leaves until there was run-off. The inoculated plants were kept in a humid chamber, at a 27 °C for 36 h. Afterwards, the plants were kept in the greenhouse for 10 days when the evaluations were completed. Three replicates were used for each isolate, with one plant considered a replicate. The pathogenicity of the isolates was confirmed by the presence of typical anthracnose symptoms. The fungal pathogen was reisolated, fulfilling Koch’s Postulates. Aggressiveness, defined by the speed at which symptoms appeared, was determined on the second day after inoculation, when the first symptoms of the disease appeared on the leaves. It was measured by the % infected leaf area and the % infected leaves. Then, analysis of variance and the Tukey test were performed using the SISVAR software program [42]. Virulence was evaluated the 2nd day after inoculation and the 12th day. We evaluated the % of infected leaves and infected leaf area according to the following scale: 0 = healthy plant; 1 = <1%, 3 = 1–5%, 5 = 6–25%, 7 = 26–50% and 9 = >50% of the diseased leaf area [43]. Virulence, which corresponds to the amount of disease induced by the pathogen in the host, was evaluated by adopting the % of infected leaf area and the % of infected leaves on the 12th day of inoculation. The difference in virulence between species was evaluated by Tukey’s test, with 0.5% significance, using Sisvar v.5.7 software [42]. Normality and homogeneity tests using Package R [44] software were performed.

## 3. Results

### 3.1. Collection and Isolation

The presence of the genus *Colletotrichum* was detected in the twenty municipalities of the two states of Brazil. A total of 51 isolates with phenotypic characteristics of *Colletotrichum* were obtained from cassava with the following frequency: 42 in Pará and 9 in Tocantins States.

After reisolation in the laboratory of the samples collected and the completion of Koch’s Postulates, the pathogenicity of 44 of the inoculated isolates of *Colletotrichum* was confirmed. Based on their morphology, 30 of the proven pathogenic isolates were randomly chosen for level-specific identification through phylogenetic analysis.

### 3.2. Phylogenetic Analyses and Species Assignment

By sequencing the partial region of GAPDH and the intergenic spacer APN2/MAT-IGS of 30 *Colletotrichum* isolates from cassava leaves, we obtained 15 haplotypes (H1-H15). The GAPDH sequences revealed a total of eight haplotypes, whereas seven haplotypes were obtained from APN2/MAT-IGS sequences. BLAST searches revealed that haplotypes H1-H7 belong to *C. gloeosporioides sensu lato*, H8-H12 to *C. orchidearum s. l*., H13-H14 to *C. boninense s. l*. and H15 to *C. truncatum s. l*. Fifteen representative isolates were randomly chosen for further multilocus analysis. Isolates from cassava were distributed in seven main clades according to the multilocus analysis (Figure 2 and Figure 3). Following the GCPSR, all species were recognized as independent phylogenetic lineages.

One of the isolates was identified as *C. truncatum*, which was well-supported on both ML and BI analyses, and retrieved as monophyletic from all individual gene trees. This species was supported by high bootstrap and posterior probability values for the TUB2 and GAPDH gene trees (Figure 2). Four isolates were identified as *Colletotrichum plurivorum*, and one isolate was *Colletotrichum musicola*. Both clades were strongly supported by ML and BI analyses and were retrieved from all individual gene trees with high support, except for *C. pluviorum* on the CHS-1 tree. Two isolates were recognized as *Colletotrichum karstii*, which were nested with significant support on multilocus analyses and the individual gene trees (Figure 2).

Two isolates were nested within *Colletotrichum chrysophilum*, two isolates were grouped with *Colletotrichum fructicola* and three isolates were identified as *Colletotrichum siamense*. They were strongly supported in both ML and BI analyses. Meanwhile, *C. chrysophilum* was shown as monophyletic with high values of bootstrap and posterior probability on the APN2 and GAP-IGS gene trees, while *C. fructicola* presented high support on most individual gene trees. On the other hand, *C. siamense* was recovered from the APN2/MAT-IGS and GAP-IGS trees with strong support in both the ML and BI analyses (Figure 3).

Most species were represented by several isolates. If considering the GAPDH region, *C. plurivorum* comprised four haplotypes and 15 isolates (H8: 2; H9: 2; H10: 10; and H11: 1 isolate, respectively). Two species were represented by one haplotype each, *C. musicola* (H12: 2 isolates) and *C. trucatum* (H15: 1 isolate), whereas *C. karsti* was represented by two haplotypes (H13: 1 and H14: 1 isolate each). In contrast, the APN2/MAT-IGS region revealed that *C. siamense* comprised three haplotypes with five isolates (H5: 1; H6: 1; and H7: 3 isolates, respectively), while *C. chrysophilum* and C. *fructicola* were represented by two haplotypes each (H1: 1, and H2: 1 isolates, and H3: 1, and H4: 2 isolates, respectively).

### 3.3. Species Prevalence of Colletotrichum Isolates by Locality

A total of seven species of Colletotrichum were found in cassava crops in northern Brazil, as follows: *C. fructicola*, *C. karsti*, *C. plurivorum*, *C. musicola*, *C. siamense*, *C. truncatum* and *C. chrysophilum* in Para State and *C. plurivorum* and *C. fructicola* in Tocantins State (Figure 4). *C. plurivorum* had the highest frequency among all species (41.07%) in Pará and 80% in Tocantins States (Figure 4).

### 3.4. Aggressiveness and Virulence of Colletotrichum Manihot esculenta

(a)Aggressiveness

On the first day after inoculation, there were no visible symptoms on the leaves. For all species, symptoms appeared on the second day after inoculation, but to varying degrees. *Colletotrichum truncatum* was the most aggressive compared with the other recovered species (F = 6.85; *df*= 6; *p* = 0.0015), causing leaf lesions in 31.8% of the plant leaves, with *C. plurivorum, C fructicula* and *C. karsti* causing leaf lesions in <20% of the leaves of *M. esculenta* (Figure 5A). However, despite differences in the number of infected leaves in the plants, the infected leaf areas in each leaf were similar for all species (F = 1.09; *df* = 6; *p* = 0.245) (Figure 5B).

(b) Virulence in cassava leaves

Ten days after inoculation, the symptoms of anthracnose in cassava plants were already well-defined (Figure 6A). The species *C. chrysophilum, C. truncatum* and *C. siamense* had a greater incidence in *M. esculenta* leaves, while the other species caused <30% of infection in the leaves (F = 4.06; *df* = 6; *p* = 0.014) (Figure 6B). Similarly, *C. chrysophilum* and *C truncatum* infected more than 5% of the total leaf area, while *C fructicola* caused <1.9% of infection (F = 11.9; *d f* = 6; *p* = 0.0001) (Figure 6C).

## 4. Discussion

Our results represent the first survey of *Colletotrichum* species associated with cassava anthracnose in commercial crops in northern Brazil. A great diversity of *Colletotrichum* species was found in the northern region of Brazil, with a total of seven species in the State of Pará and two in Tocantins. After isolation, pathogenicity, virulence and phylogenetic analyses, our results show that *C. plurivorum*, *C. siamense*, *C. fructicola*, *C. chrysophilum*, *C. karsti*, *C. musicola* and *C. truncatum* were responsible for anthracnose in cassava leaves with different levels of severity of the symptoms. Among the seven species identified, *C. chrysophilum* proved to be the most virulent, while *C. fructicola* was the least. *Colletotrichum plurivorum* (*sic*) predominated in cassava crops in both Pará and Tocantins, with a higher incidence in Tocantins. A similar prevalence has been reported in other crops across the Amazon region and northeastern states of Brazil [41]. Such a prevalence may be attributed to the species’ diversity and wide host range [12,45], or to facilitated dissemination due to small-scale farming, extensive exchange of propagation material among producers and consecutive planting lacking proper cultural treatment. Although, in crops in tropical countries, this species is reported as the main cause of anthracnose [46], including that in okra (*Abelmoschus esculentus*) in Para State, Brazil [47], the high prevalence of *C. plurivorum* found in our study is not necessarily causing significant virulence in cassava. It could be that this species of *C. plurivorum* is less aggressive than others or is the most well-tolerated by the cassava var. Formosa.

*Colletotrichum fructicola* was also found both in the Pará and Tocantins States. This species is well-distributed in several regions of Brazil, being reported different ecosystems [14,48]. Although *C. fructicola* is associated with anthracnose disease on several fruit crops [49,50,51,52] and others including cassava [14,22], our study revealed that this species is less aggressive and virulent than others in cassava plants var. Formosa. On the other hand, *C. chrysophilum* was found only in Pará State crops and with a lower prevalence; however, it was the most aggressive and virulent species in cassava crops. Our results are in accordance with those of another study [23], which reported this species in Brazil as the anthracnose agent in cassava crops. Similarly, this disease in blueberry plants [53], açaí plants [54] and banana fruits [55] was found to be caused by *C. chrysophilum*. Therefore, our findings could sound an alarm that there is a need for the establishment of preventive measures for this pathogen. The other species found here, *C. karsti*, *C. musicola*, *C. siamense* and *C. truncatum*, also caused anthracnose in cassava leaves but had a virulence level 3, i.e., 1–5% of leaf area affected, less than that of *C. chrysophilum*. However, these findings should nonetheless serve as an alert for farmers because these species have been highlighted as causes of anthracnose on others crops. For instance, *C. karsti* was reported in different crops in Brazil [56,57,58], along with *C. musicola* [59], *C. siamense* [60,61,62] and *C. truncatum* [63], and those crops could serve as reservoir hosts. There is a potential risk that when new clones of cassava different to var. Formosa are introduced, that they could be affected differently by the different species of *Colletotrichum*. When analyzing the results of this study and also taking into account that var. Formosa is the most cultivated cassava in the sampled producing regions of Tocantins and Pará, and considering the potential for pathogenic variability of *Colletotrichum* spp., it can be surmised that there is a differential interaction in response to infection for this genotype, which is revealed in aggressiveness and virulence bioassays. Therefore, as a control strategy, producers must continue planting different clones and other cassava cultivars, to avoid large epidemics of anthracnose in cultivated areas. It is known that for pathogens with high variability, the planting of one or a few cultivars in large areas stimulates the rapid breakdown of resistance, resulting in strong epidemics.

In conclusion, knowledge of the prevalence of pathogenic species, their aggressiveness and their virulence can be important information in the adoption of control measures, such as crop rotation and the use of resistant cultivars. Mapping the prevalence of these species and their virulence in each producing region can contribute to the development of genes providing resistance to pathotypes and thus raise the durability of the resistance of cultivars. There should also be a search for new sources of resistance that are more effective than those at present in controlling the disease. These measures, in addition to prolonging the useful life of cultivars, will also serve to reduce production costs, avoid high environmental risks due to the excessive use of pesticides and contribute to increasing the productivity of the cassava crop.

## Figures and Tables

**Figure 1 jof-10-00367-f001:**
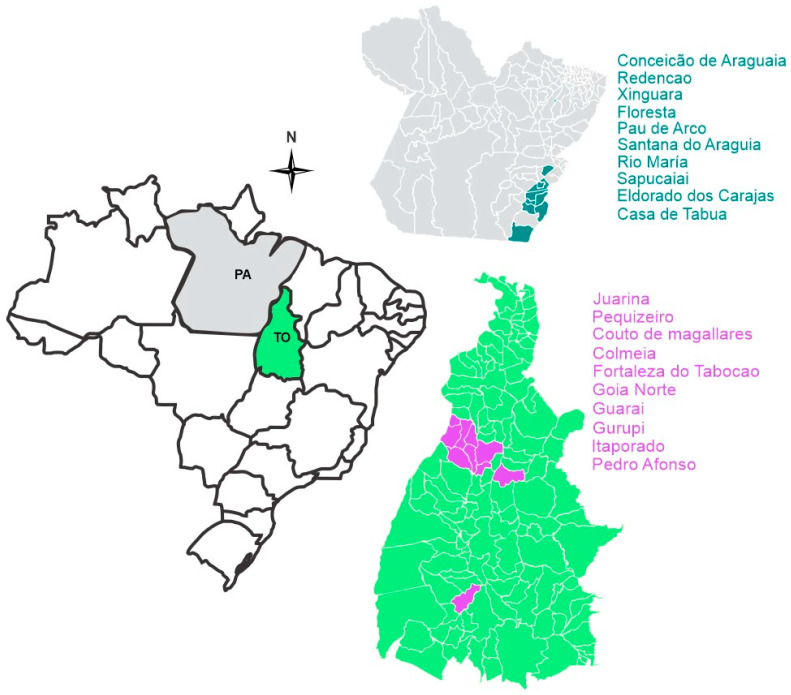
Locations (green and purple) of samples collected in the Pará (gray) and Tocantins (green) states of cassava crops infested with *Colletotrichum* species.

**Figure 2 jof-10-00367-f002:**
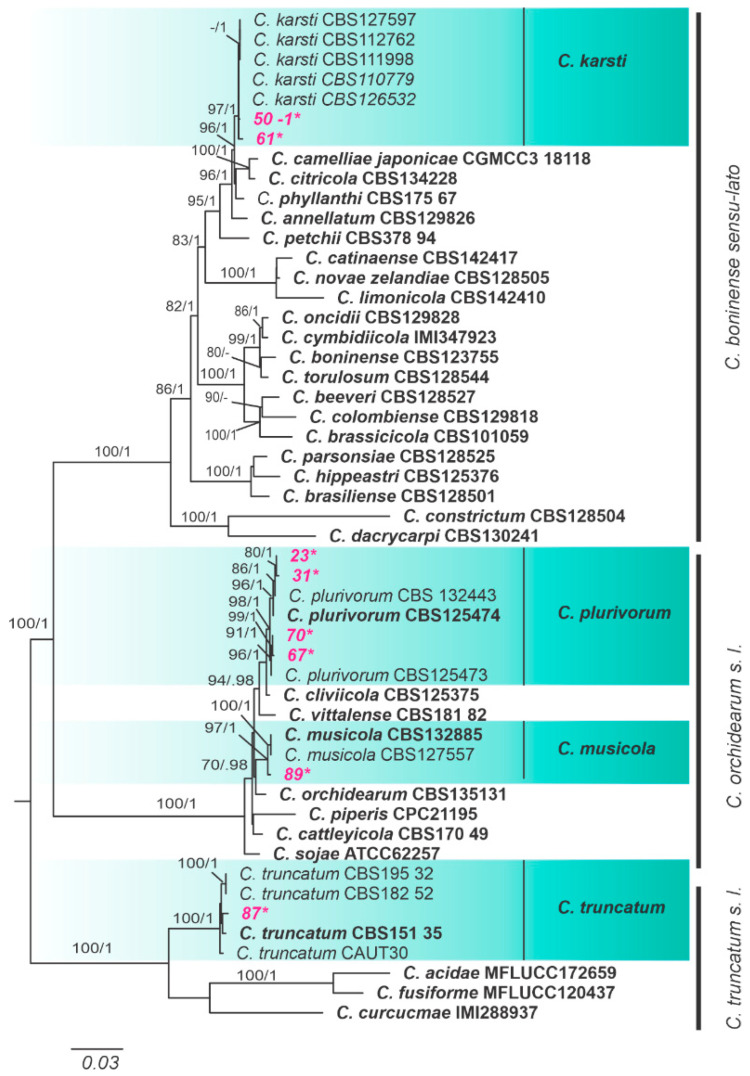
Maximum likelihood tree of *Colletotrichum* species inferred from a concatenated alignment of the following genes: ACT, TUB2, GAPDH, CHS-1 and HIS3. Bootstrap support values (ML ≥ 70) and Bayesian posterior probability values (PP ≥ 0.95) are shown at nodes. “-” indicates non-significant support or node absence, former types are emphasized in bold Cassava isolates from the present study are highlighted in pink and marked with asterisks. The tree is rooted at the midpoint. The scale bar indicates the estimated number of replacements per site.

**Figure 3 jof-10-00367-f003:**
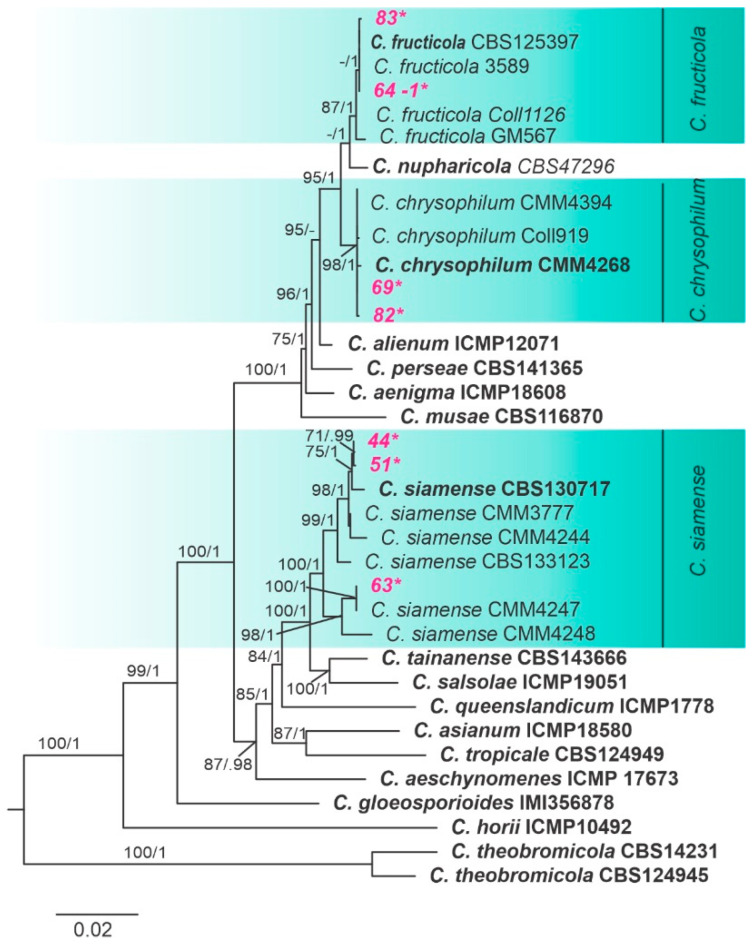
Maximum likelihood tree of the *Colletotrichum gloeosporioides* species complex inferred from a concatenated alignment of Apn2/MAT-IGS, Apn2 and GAP2-IGS. Bootstrap support values (ML ≥ 70) and Bayesian posterior probability values (PP ≥ 0.95) are shown at nodes. “-” indicates non-significant support or node absence, former types are emphasized in bold. Cassava isolates from the present study are highlighted in pink and marked with asterisks. The tree is rooted at the midpoint. The scale bar indicates the estimated number of replacements per site.

**Figure 4 jof-10-00367-f004:**
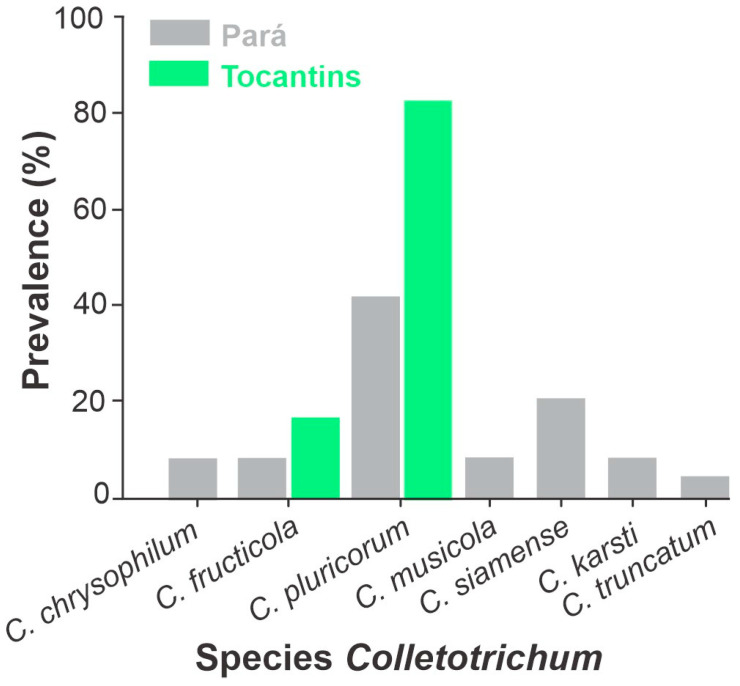
Prevalence (%) of *Colletotrichum* species in cassava leaves collected from commercial areas in the states of Pará and Tocantins.

**Figure 5 jof-10-00367-f005:**
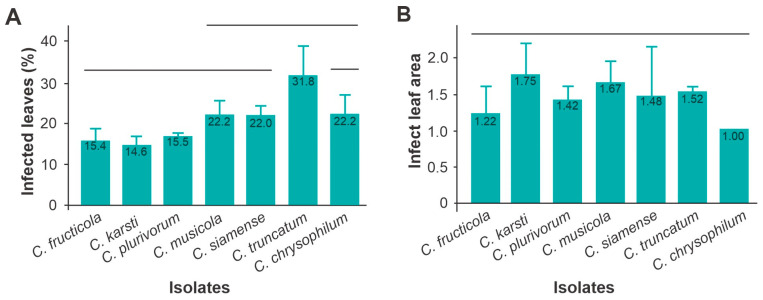
Aggressiveness of *Colletotrichum* species on cassava plants, var. Formosa, at 2 days after inoculation. Incidence or percentage of infected leaves (**A**) and infected leaf area (**B**). Vertical bars grouped under the same horizontal line are not significantly different (Tukey < 0.05).

**Figure 6 jof-10-00367-f006:**
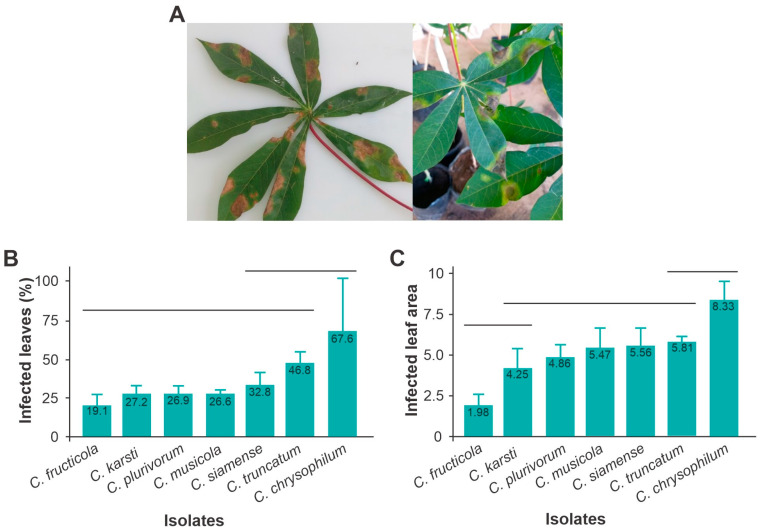
Symptoms of anthracnose on leaves in the cassava plant, var. Formosa, on the 10th day after inoculation with *Colletotrichum* (**A**). Incidence or percentage of infected leaves (**B**) and infected leaf area (**C**). Vertical bars grouped under the same horizontal line are not significantly different (Tukey test < 0.05).

## Data Availability

Data are contained within the article.

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
