# Peer review of "Diversity, Prevalence and Virulence of *Colletotrichum* Species Causing Anthracnose on Cassava Leaves in the Northern Region of Brazil"

_jof, 2024, doi:10.3390/jof10060367_

Round 1

Reviewer 1 Report

This work provides a study of the diversity, prevalence and virulence of Colletotrichum species, which caused anthracnose on cassava leaves in the northern region of Brazil. The manuscript might be publishable, but the current text needs to be improved. 

1.      The references cited in the text should be arranged in chronological order.

2.      The number of the isolates are not very clear for both the total and haplotypes. Please check and make it clear.

3.      In the phylogenetic analyses, outgroups are lack for each analysis. Despite all the isolates are Colletotrichum, strains in other complex or other genus should be chosen as outgroup to make sure the reliability for the analyses. And the root of the tree is not appropriate in the middle.

4.      The figures and tables of the levels of pathogenicity test, virulence and so on should be provided.

5.      There are many writing and punctuation mark mistakes. That should not appear in the submission manuscript. The authors should be more conscientious for your work.

Author Response

Dear Review

We are pleased to re-submit our manuscript jof-2988555 entitled “Diversity, Prevalence and Virulence of Colletotrichum Species Causing Anthracnose on Cassava Leaves in the Northern Region of Brazil” to the Journal of Fungi. First of all, we would like to thank you for reviewing our manuscript and pointing out areas for scientific improvement. We have gladly taken them all into account in the revised version of our manuscript. To make it easier to check the changes, the lines of the manuscript have been numbered and all changes have been marked in red in the body of the manuscript for your information.

We inform you that the English has been improved by a professional and has also been checked by the software (www.grammarly.com) to address the concerns raised. Finally, if you have any further questions about the current version, please let us know and we'll be happy to answer them.

Sincerely,

Prof. Dr. Gil Rodrígues dos Santos

Production Vegetal Pos-graduate Program 

Federal University of Tocantins

Tocantins – Brazil

Reviewer 2 Report

This paper isolated many Colletotrichum strains from cassava leaves cultured in the northern region of Brazil, and analyzed their diversity, prevalence and virulence. This paper expands our knowledge of the distribution and diversity of Colletotrichum in the world. However, several detailed information is missing, and the authors should add them as mentioned below.

Please add a table to Materials and Methods to list the strains isolated in this study (including information, such as strain name, collection place, and so on).

It is highly recommended to provide representative disease pictures for pathogenic inoculation analysis.

P5, The number of isolated strains is confusing. “A total of 30 isolates with phenotypic characteristics”, “42 in Pará and 9 in Tocantins States”. “the pathogenicity of 44 of the inoculated isolates of Colletotrichum was confirmed”. So, 30, 42+9, or 44?

P5, “Phylogenetic analyses and species assignment”. The subtitle appears to be missing the serial number "3.2.".

P5, “ten isolates (H1-H7) were similar to sequences”. In my mind, H1-H7 contains only 7 isolates.  “17 isolates (H8-H12) similar to C. orchidearum s. lat”. H8-H12 contains 17 isolates?

P5, Colletotrichum were obtained were obtained from”.

P8, the data in Figure 4A are inconsistent with the data in Figure 4B. In 4A, C. chrysophilum is the lowest; however, in 4B, C. truncatum is the lowest. Why?

According to the early or late infection process, the author divides pathogens into two phenotypes: Aggressiveness and Virulence. Please add information to explain the reasons for this distinction.

P8-P9, both Figure 5 and Figure 6 perform significance analysis in the main text. It is recommended to mark the significance on the figures.

“Table Suplementar S2. Colletotrichum spp. Strain estudied, detalling collection and access GenBank.” There are several grammatical errors in this table title.

Author Response

(The authors gave the same response as above.)

Reviewer 3 Report

Authors described well the Colletotrichum funghi collection isolated from Cassava plants. Moderate English check is required. There is a lot of information collected from experimental sites. Paper could be improved by adding the information about the possibilities of chemical control and its treshold levels for the Cassava plants.

In the introduction section, could You please add some information about the harmfulness of the Colletotrichum species for Cassava plants? What are the maximum and average losses in crop production (if any)? What is the epidemiology of the diseaese? Which parts of the plants can be infected? Only leaves? Could You please add some information about the mentioned topic and the pottential source of the causal agents of the diesease?

In the M&M section could You please describe the metohodology of selecting the experimental sites? There is an information about "five properties with M. esculenta in different stages of growth and maintained with family labor". Were they selected randomly? Why 5?

In the whole paper Could You please check the names of the Colletotrichum species? In the Disscusion section there is Colletotrich plurivorum and Colletotrichm fructicola. Please give these funghi a proper names.

Did You try to examine the efficacy of main gropus of fungicides in the control of mycellium growth on PDA plates of the isolated species? Maybe it could be done in the near future to enrich the paper or to continue the research in this area.

Author Response

(The authors gave the same response as above.)

Reviewer 4 Report

I find this article interesting and sure it will provide useful information for cassava growers in Brazil and everywhere the crop is commercially used. However, I have some concerns,  comments and recommendations:

From the abstract and introduction, we have that"Our findings provide accurate identifications of species of Colletotrichum causing anthracnose in cassava crops and provide important information for a more effective control of this disease and the adoption of anthracnose management strategies. These include the development of cultivars with resistance genes to the different pathotypes prevalent in the areas of cultivation, which will potentially reduce production costs". The authors do not explain how the information they uncovered will lead to a more effective control of the disease; I find expressions like this to be kind of a cliché. Please provide some clues as to how to use the new information provided for you in effective control measures of the disease, or rephrase what you have said regarding this matter.

- Could "Prevalence of Colletotrichum species" be better written like Species prevalence of Colletotrichum isolates by locality? I find confusing the use of the term "prevalence" by the authors. Regarding the use of the term prevalence, is this sentence correct: all seven species were prevalent, with C. plurivorum being the most prevalent in both states?

- I find problematic the use of other terms very specific to plant pathology: for example, on page 8, what do they mean by "more anthracnose"? In a few other instances, the authors succumb to the temptation of being a little bit colloquial.

Finally, the main sentence of the Discussion part is the most difficult to follow of all: "Thus, of the seven species found the C. chrysophilum and C. fructicola were the most and least virulent, respectively. Colletotrich plurivorum (sic) was the most prevalent species in cassava crops in both in Pará and Tocantins States with highest incidence in the Tocantins, Smilarly in other crops in the Amazon region and North-eastern states in Brazil has been reported (Cavalcante et al., 2019); This prevalence can be associated with diversity and large hots range for this species (Damm et al., 2018; Liu et al., 2019) or the easy dissemination favored by the presence of several small crops in the majority of properties, great exchange of propagation material among producers, successive plantings and conducted without the proper cultural treatments." This whole paragraph, which should summarize the article's findings, is so obscure that nothing is clearly stated in the end. I would recommend rewriting the entire Discussion part: rather than its content, it is the problematic wording that is the issue at hand.

- Some sentences are truncated, while other expressions are repeated in tandem and few spells of specific epithets are employed throughout the article.

- I would prefer if the authors use the term tropical countries instead of developing countries when talking about the regions of the world where cassava is cultivated. For example, we find cassava in most Latin American countries except in those countries of the region where the climate is not tropical. In other places of the world, there are countries which are considered developed AND tropical and cassava is also cultivated. In the same vein, I believe it is inappropriate to say "in tropical rainforest, tropical dry savanna and tropical wet savanna climates" since they are not climates but ecosystems (with a particular climate, which is also true).

- May the authors change the word "property" for plot, location, parcel, farm, sampling sites or something more easily relatable to what they mean?

- Is it color really necessary for Figure 1?

Author Response

(The authors gave the same response as above.)

Round 2

Reviewer 1 Report

The authors didn't response well for the comments 3 and 4. You need to indicate which isolate is outgroup in the manuscript, as well as the legends of figures. The photos showed the results of the experiments for pathogenicity test should be provided as a figure.

Besiades, I didn't see any corrections for the manuscript which revised in the last version.

The authors didn't response well for the comments 3 and 4. You need to indicate which isolate is outgroup in the manuscript, as well as the legends of figures. The photos showed the results of the experiments for pathogenicity test should be provided as a figure.

Besiades, I didn't see any corrections for the manuscript which revised in the last version.

Author Response

May 07, 2024

Dear Review

We are pleased to re-submit our manuscript jof-2988555 entitled “Diversity, Prevalence and Virulence of Colletotrichum Species Causing Anthracnose on Cassava Leaves in the Northern Region of Brazil” to the Journal of Fungi. First of all, we would like to thank you for reviewing our manuscript and pointing out areas for scientific improvement. We have gladly taken them all into account in the revised version of our manuscript. To make it easier to check the changes, the lines of the manuscript have been numbered and all changes have been marked in red in the body of the manuscript for your information.

Besides, we thank for his comment and apologize if the new version has not been sent to you. We have now attached the new version of our work with the suggestions already made and included in the manuscript.

Sincerely,

Gil Rodrigues dos Santos

(Corresponding author)

Reviewer 2 Report

This paper isolated many Colletotrichum strains from cassava leaves cultured in the northern region of Brazil, and analyzed their diversity, prevalence and virulence. More detailed information was added and the revision is good. The current version is suitable for publication.

Figure 6 is missing a title.

Author Response

May 07, 2024

Dear Review

We are pleased to re-submit our manuscript jof-2988555 entitled “Diversity, Prevalence and Virulence of Colletotrichum Species Causing Anthracnose on Cassava Leaves in the Northern Region of Brazil” to the Journal of Fungi. First of all, we would like to thank you for reviewing our manuscript and pointing out areas for scientific improvement. We have gladly taken them all into account in the revised version of our manuscript. To make it easier to check the changes, the lines of the manuscript have been numbered and all changes have been marked in red in the body of the manuscript for your information.

Sincerely,

Gil Rodrigues dos Santos

(Corresponding author)
